# Social support and physical exercise: The mediating role of achievement emotions, and the moderating role of exercise motivation and feedback literacy

Hui Zhang[1,2]*, Yuzhong Zhang[1], Yi Zhou[3], Baichao Xu[4]

**1** Teaching Department of Public Courses, Guangdong Songshan Polytechnic, Shaoguan, Guangdong, China, **2** Hainan Provincial Key Laboratory of Sports and Health Promotion, Key Laboratory of Emergency and Trauma, Ministry of Education, The First Affiliated Hospital of Hainan Medical University, Hainan Medical University, Haikou, China, **3** General Services Department, Guangdong Songshan Polytechnic, Shaoguan, Guangdong, China, **4** Department of Physical Education, Hainan Medical University, Haikou, Hainan, China

* zhshg2024@163.com

## Abstract

### Objective

While research on physical exercise has predominantly focused on rational decision-making behaviors related to health benefits, comparatively less attention has been paid to achievement emotion and its role in promoting physical exercise. Building upon dual-system theory (DST), this study examines whether achievement emotion mediates between social support and physical exercise, integrating expected value and information processing theories. It also explores how this mediation effect depends on sports motivation and feedback literacy.

### Methods

A questionnaire survey was conducted among 534 college students (281 male, 253 female). Data were analysed using hierarchical regression procedures in terms of moderated mediation.

### Results

The results show that achievement emotion partially mediates the relationship between social support and physical exercise. Specifically, the interaction of social support × sports motivation indicates that the strength of the relationship between social support and achievement emotion increases linearly with higher levels of sports motivation; similarly, the interaction of social support × feedback literacy indicates that the strength of the relationship between social support and physical exercise also increases linearly through feedback literacy level. Finally, the interaction of

**Data availability statement:** Raw data for all statistical analyses are provided in S1 Achievement emotion in social support and physical exercise Dataset. http://datadryad.org/share/-enw9vXLoz97M58FUveSHpBFy_X_agYLbMLM52DVZS8

**Funding:** This work was supported by the [Hainan Provincial Key Laboratory of Sports and Health Promotion Project] under Grant [number HNYJ2023005]. The project that facilitated data acquisition and the use of analytical software supported this study.

**Competing interests:** No potential conflict of interest was reported by the author(s).

achievement emotion × feedback literacy indicates that the strength of the relationship between achievement emotion and physical exercise increases linearly through feedback literacy level.

## Conclusions

These findings suggest that attention should be paid to achievement emotion, sports motivation, and feedback literacy when designing social support interventions to promote physical exercise. This research provides a theoretical reference for improving levels of physical exercise among college students.

## 1.  Introduction

Physical activity is a key factor for improving physical fitness [1] and a necessary component of supporting health [2] and promoting sustainable human development [3,4]. Studies in China and globally have shown that although physical exercise has many benefits, few college students engage in regular physical exercise, and there is a notable absence of high-intensity exercise among these students [5,6]. To deal with the problem of insufficient physical exercise, the Chinese government has introduced a series of policies aimed at encouraging college students to be more active. In 2021, various governmental departments proposed measures to improve PE in schools, including ensuring the provision of teachers, venues, and sufficient class hours, with the goal of increasing the proportion of individuals who regularly participate in physical exercise to 38.5% by 2025. However, a 2023 survey by the China Youth Network on college students' physical exercise indicated that 50% of college students exercise fewer than three times per week. Although college students generally have recognize the importance to exercise, the frequency and intensity of their actual exercise behaviors remain inadequate [7]. This may be due to a greater need to focus on the emotional experience of physical exercise, rather than solely its health benefits [7,8], which can be better understood through the integration of multiple theoretical models.

The dual-system theory (DST) was first proposed by Reber [9]. DST suggests that human information processing is divided into two systems with distinct properties. System 1 is fast and intuitive, relying on intuition and emotion, while System 2 is slow and rational, grounded in modern analysis and logical reasoning [10]. As society evolves, human beings have increasingly entered an era focused on emotional experience [7], highlighting the need to consider emotional experiences when addressing the effect of physical exercise [11]. Achievement emotions refer to emotions that are directly related to achievement activities (such as physical exercise) or the outcomes of achievement (success and failure), and can fully reflect the psychological development process of students during their physical exercise [2]. However, previous research on physical activity predominantly emphasized the dominance of DST's System 2, focusing primarily on the health benefits of exercise [7]. This study, therefore, focuses on DST's System 1 and its interaction with System 2, combining theories of expected value, social capital, and information processing ability to explore

how an emotion-centered model affects physical exercise. This study will contribute to uncovering the underlying psychological mechanisms through which social support influences college students' physical exercise and provide a theoretical basis for addressing the challenges in promoting regular physical activity among this population.

## 2. Literature review and hypotheses establishment

### 2.1. Social support, achievement emotion, and physical exercise

As previously highlighted, physical exercise can alleviate depression and anxiety, as well as improve well-being and social adjustment. A study [7] emphasized that emotion is the key to understanding behavioral performance and plays an important role in promoting physical exercise. According to DST, human behavior arises from the interaction between System 1 and System 2 [10], with System 1 (emotional experience) being the dominant driver [8].

A longitudinal study concluded that social support for exercise was positively correlated with physical exercise, although this effect was influenced by race, gender, and type of support [12]. Steptoe et al. [13] reported that college students with lower social support consumed more alcohol, smoked more, and had lower levels of physical exercise, whereas higher social support was associated with greater levels of physical exercise.

The theory of social capital also helps to explain the relationship between social support and physical exercise. Bourdieu's theory puts forward two concepts, field and capital, and emphasizes the role of various social relationships in the formation of networks. Support from social networks such as family, friends, and colleagues is a key resource for initiating and sustaining physical exercise [14]. A study of European and African American women [15] found a significant positive correlation between physical exercise, social support, and mood, with physical exercise and social support serving as positive predictors of mood. Another study of Korean immigrants in the United States [16] also showed that social support fostered positive emotions that affected well-being and mental health. However, a study from China [17] reported that support from teacher autonomy support positively predicted physical activity in college students, mainly through the classroom enjoyment activities.

Although the positive relationship between social support, achievement emotion, and physical exercise has been clarified, studies have generally focused on how physical exercise and social support can predict emotion, while the influence of emotion on physical exercise has rarely been considered. Indeed, although there have been studies about the influence of social support and achievement emotion on physical exercise, such studies are relatively few and lack depth. It is, therefore, necessary to further verify the predictive effects of social support and achievement emotion on physical exercise, especially among Chinese college students.

### 2.2. Sports motivation as a regulator

Previous studies have demonstrated the correlation between social support, motivation, and emotion [18]. DeFreese et al. [19] conducted a survey of American college athletes and found that teammate support played a key role in fatigue and motivation during sports activities. According to Tur-Porcar, et al. [20], sports motivation and emotional regulation play an important role in generating exercise commitment in sports organizations.

These findings are consistent with other research on expected value theory. Eccles and Wigfield suggested that an individual's motivation was determined by his or her expectation of success and subjective value of the task. When expectations and perceived value are high, individuals are more likely to have positive emotions. Izni et al. [21] also found a significant correlation between expected value and social support and sports motivation. However, the relationship between social support, motivation, and emotion appears to be influenced by cultural differences. Europeans and Americans experience heightened emotions when receiving social support, while the opposite effect has been observed among Japanese individuals [18]. Additionally, Chinese studies have explored the relationship between social support and sports motivation [22], as well as sports motivation and mood [23]; however, the role of motivation in regulating the interplay between social support and emotion requires further verification.

## 2.3. Feedback literacy as a regulator

The concept of feedback literacy, first proposed by Carless and Boud, refers to students' ability to understand and use feedback information effectively. This study was conducted based on the above-mentioned definitions. This behavior is highly regarded by the educational community. According to information processing theory, incorporating feedback improves confidence in both individual and group decision-making [24]. Research has shown that physical exercise levels increase under feedback conditions compared to situations without feedback [25]. Feedback thus appears to be closely related to people's decision-making and behavior. Irandoust et al. [26] reported the positive impact of feedback on the physical exercise behavior of corporate executives, and increasing positive feedback and peer support helped increase their levels of physical exercise [27]. The effect of social support on physical exercise is thus influenced by feedback. According to emotion theory, conscious emotion evaluates behavior through feedback, and the influence of emotion on behavior is indirect [28]. Physical exercise, as a behavior, should be influenced by emotions through feedback. However, due to the limited attention from researchers, feedback literacy has mainly been studied in Chinese, English, and other subjects, but not for physical exercise. This raises the question: Does feedback literacy have the same predictive effect on physical exercise as feedback? Whether feedback literacy regulates the relationship between achievement emotion and physical exercise is therefore worth studying. Based on the above discussion, the following hypotheses are proposed (see Fig 1):

H1. Achievement emotion has a positive mediating effect between social support and physical exercise.

H2. Sports motivation regulates the relationship between social support and achievement emotion.

H3. Feedback literacy regulates the relationship between social support and physical exercise.

H4. Feedback literacy regulates the relationship between achievement emotion and physical exercise.

## 3. Methods

### 3.1. Participants

In this study, the convenience sampling method was used to select students from Guangdong Songshan Polytechnic and Hubei Enshi College. Before the class, the PE teacher explained the purpose of the research project and provided information about the anonymity of the responses and other study characteristics. This study was conducted in compliance with the

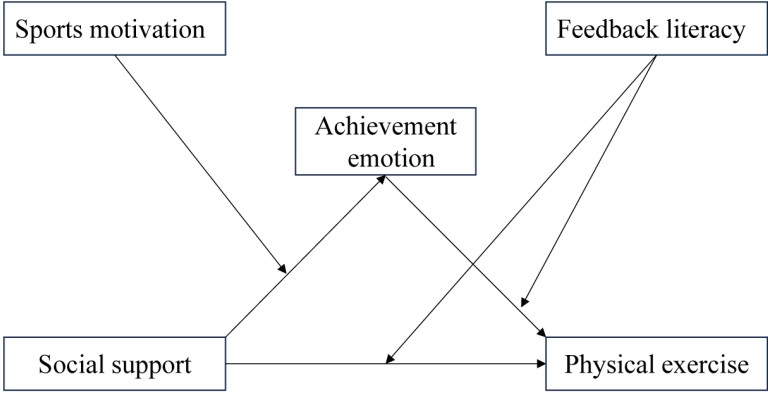

**Fig 1. Proposed moderated mediation model.**

principles outlined in the Declaration of Helsinki. This study was approved by the Ethics Approval Panel of the Project Approval Committee. After obtaining written informed consent, questionnaires were sent to them through the questionnaire star platform. As the study did not involve any minors, parental or guardian consent was not required. After students completed the questionnaire, there was a lottery draw, and some students received a reward of 1–5 yuan. In this study, Raosoft was used to calculate the minimum sample size, which was determined to be 380. In total, 646 questionnaires were returned. Of these, 110 questionnaires were discarded due to dual responses or outliers, leaving 534 valid questionnaires. Of the valid responses, 281 (52.6%) were from male students and 253 (47.4%) from female students. There were 411 first-year students (77%), 68 sophomores (12.7%), and 55 juniors (10.3%). By institution, there were 241 undergraduates (45.9%) and 293 junior college students (54.8%). The recruitment period for this study began on 25/04/2024 and concluded on 17/05/2024.

### 3.2. Measures

Social support was measured using the Perceived Social Support Scale (PSSS) [29]. The scaleconsists of 12 questions. Items were rated using a 7-point Likert scale (1 = strongly disagree, 7 = strongly agree). The higher the mean, the higher the self-perceived social support. The overall item reliability coefficient of this scale was 0.972.

Sports motivation was measured using the Sports Motivation Scale (SMS-II) from the SMS revised by Pelletier [30]. The scaleconsists of 18 questions. Items were rated using a 7-point Likert scale (1 = strongly disagree, 7 = strongly agree). The higher the mean, the higher the motivation. The overall item reliability coefficient of this scale was 0.960.

Achievement emotion was measured using the Achievement Emotions Questionnaire (AEQ) by Pekrun [31]. The scaleconsists of 28 questions. Items were rated using a 7-point Likert scale (1 = strongly disagree, 7 = strongly agree). The five negative emotions (e.g., anxiety and anger) were scored in reverse, and the higher the average value after adding the positive emotions, the higher the level of emotion related to PE classroom achievement. The overall item reliability coefficient of this scale was 0.958.

Measurement of feedback literacy used the Feedback Literacy Behavior Scale (FLBS) developed by Dawson [32]. The scaleconsists of 24 questions. Items were rated using a 6-point Likert scale (1 = strongly disagree, 6 = strongly agree). The higher the average, the higher the feedback literacy. The overall item reliability coefficient of this scale was 0.977.

Physical exercise was measured using Liang Qingde's [33] Physical Exercise Rating Scale, which consisted of three items: exercise intensity, frequency, and time. Each item is scored on a scale of 1–5. Physical exercise amount = intensity × (time − 1) × frequency; the higher the score, the higher the level of physical exercise. The overall item reliability coefficient of this scale was 0.675.

### 3.3. Data analysis

SPSS 26 and the SPSS PROCESS macro were used to analyze the collected data. First, descriptive statistics of the study variables were used to determine whether the questionnaire data conformed to the hypothesis of multivariate normality. Second, internal consistency analysis was performed for a single dimension of each variable to estimate the Cronbach's α coefficient. Third, the degree of correlation between all variables was assessed through bivariate correlation analysis. Fourth, simple regression analysis was conducted using the PROCESS macro to verify the mediating effect of the proposed achievement emotion, and PROCESS model 28 was used to test the interaction effect of two moderating variables (social support × sports motivation, achievement emotion × feedback literacy, social support × feedback literacy). If the interaction effect was significant, a deviation score of ±1.0 centered on the mean of each variable was used as the criterion to explain the regression slope between the high and low groups. All statistical significance was tested at λ = 0.05.

## 4. Results

### 4.1. Common method bias

To control for common method bias, the PE teacher involved in distributing the questionnaires were uniformly trained. All students participated voluntarily and completed the questionnaires anonymously. The Harman's single-factor test was

used to examine common method bias. Upon examination, the questionnaire yielded 13 factors, with the first common factor explaining 37.33% of the variance, which is less than 40%. There is no significant common method bias in the questionnaire.

## 4.2. Correlation analysis

The correlation coefficients and descriptive statistical results among all observed variables are shown in Table 1. The correlation coefficients between all predictors ranged from 0.158 to 0.707, which indicates a significant positive association between them. Specifically, from smallest to largest, the correlations between physical exercise and feedback literacy, achievement emotion, social support, and sports motivation were in the range of $r = 0.158–0.250$.

## 4.3. Hierarchical regression

### 4.3.1. Simple mediation effect.
The regression coefficients of the path from social support to achievement emotion ($\beta = .222$, $p = .000$), between achievement emotion and physical exercise ($\beta = 4.311$, $p = .003$), between social support and physical exercise ($\beta = 2.582$, $p = .002$), and between social support and physical exercise ($\beta = 2.582$, $p = .002$) are shown in Table 2. The regression coefficient between social support and physical exercise in the total effect model ($\beta = 3.540$, $p = 0.000$) was statistically significant. The indirect effect of social support on physical exercise was significant through achievement emotion ($\beta = .958$, CI [.209, 1.818], $p = .005$). According to the mediated effect size formula proposed by MacKinnon et al. (2007): (total effect − direct effect)/ (total effect), the mediated effect size of achievement emotion was about 22.2% (.563 − .438/.563 = .271).

### 4.3.2. Adjustment effect.
As can be seen from Table 3, the interaction between social support and sports motivation significantly moderates the prediction of achievement emotion (achievement emotion: B = −0.132, t = −6.163, p < 0.001). This means that sports motivation not only plays a moderating role in the relationship between social support and achievement emotion, but also moderates the influence of social support on achievement emotion. The interaction between social support and feedback literacy was significant in predicting exercise behavior, where it also played a significant moderating role (physical exercise: B = 2.061, t = 2.304, p = 0.022). This means that feedback literacy not only moderates the direct prediction of physical exercise from social support, but can also regulate the influence of social support on physical exercise. Similarly, the interaction between achievement emotion and feedback literacy is also significant in predicting exercise behavior. The interaction terms for emotional experience and volitional control significantly moderated the prediction of physical exercise (physical exercise: B = −3.628, t = −2.638, p = 0.009). This means that feedback literacy not only regulates the direct prediction of physical exercise from achievement emotion, but can also regulate the influence of emotional experience on exercise behavior.

In addition to examining the interaction of social support × sports motivation on achievement emotion and the interaction of social support × feedback literacy and achievement emotion × feedback literacy on physical exercise, the regression

**Table 1. Correlations among variables.**

| Variables | 1 | 2 | 3 | 4 | 5 | M | SD |
|---|---|---|---|---|---|---|---|
| 1. Social support | 1 | | | | | 4.920 | 1.145 |
| 2. Sprots motivation | .707** | 1 | | | | 4.686 | 1.048 |
| 3. Achievement emotion | .386** | .341** | 1 | | | 3.508 | .659 |
| 4. Feedback literacy | .578** | .601** | .378** | 1 | | 4.334 | .945 |
| 5. Physical exercise | .194** | .250** | .191** | .158** | 1 | 19.720 | 20.917 |

Note:

**p < .01

**Table 2. Mediating effect of achievement emotion between social support and physical exercise.**

| Path | β | S.E. | t | P |
|---|---|---|---|---|
| Social support → Achievement emotion (a) | .222 | .023 | 9.658 | .000 |
| Achievement emotion → Physical exercise (b) | 4.311 | 1.452 | 2.969 | .003 |
| Social support → Physical exercise (c´) | 2.582 | .836 | 3.089 | .002 |
| Social support → Physical exercise (c) | 3.540 | .777 | 4.558 | .000 |
| | Effect | S.E. | z | p |
| Indirect effect and significance using normal theory test | | | | |
| Sobel test | .958 | .339 | 2.824 | .005 |
| Bootstrap results for indirect effect | Effect | S.E. | LL 95% BC | UL 95% BC |
| | .958 | .404 | .209 | 1.818 |

Note: Bootstrap sample size = 1000, LL = Lower level, UL = Upper level, BC = Bias-corrected confidence intervals, c´ = direct effect

**Table 3. Regression analysis of mediated models.**

| Model | β | S.E. | t | P | LLCI | ULCI |
|---|---|---|---|---|---|---|
| Outcome variable: Achievement emotion ($R^2$ = .215, F = 48.308, p = .000) (Constant) | .112 | .031 | 3.581 | .000 | .050 | .173 |
| Social support | .176 | .031 | 5.623 | .000 | .115 | .238 |
| Sports motivation | .177 | .037 | 4.741 | .000 | .104 | .250 |
| Social support × Sports motivation | −.132 | .021 | −6.163 | .000 | −.174 | −.090 |
| Outcome variable: Physical exercise ($R^2$ = .070, F = 7.932, p = .000) (Constant) | 19.283 | 1.020 | 18.897 | .000 | 17.278 | 21.287 |
| Social support | 1.277 | 1.040 | 1.228 | .220 | −.766 | 3.320 |
| Achievement emotion | 5.668 | 1.568 | 3.614 | .000 | 2.587 | 8.749 |
| Feedback literacy | .397 | 1.176 | .338 | .736 | −1.914 | 2.708 |
| Social support × Feedback literacy | 2.061 | .894 | 2.304 | .022 | .304 | 3.817 |
| Achievement emotion × Feedback literacy | −3.628 | 1.375 | −2.638 | .009 | −6.329 | −.926 |
| Conditional effect of predictor at values of moderator variables Moderator: Sports motivation | Effect | S.E. | t | P | LLCI | ULCI |
| −1SD | .314 | .039 | 7.979 | .000 | .237 | .392 |
| Average | .176 | .031 | 5.623 | .000 | .115 | .238 |
| +1SD | .038 | .038 | 1.014 | .311 | −.036 | .112 |
| Moderator: Feedback literacy (Social support × Feedback literacy) | Effect | S.E. | t | P | LLCI | ULCI |
| −1SD | −.670 | 1.567 | −.427 | .669 | −3.749 | 2.409 |
| Average | 1.277 | 1.040 | 1.228 | .220 | −.776 | 3.320 |
| +1SD | .945 | 3.224 | 1.065 | .003 | 1.132 | 5.317 |
| Moderator: Feedback literacy (Achievement emotion × Feedback literacy) | Effect | S.E. | t | P | LLCI | ULCI |
| −1SD | 9.096 | 2.319 | 3.923 | .000 | 4.541 | 13.651 |
| Average | 5.668 | 1.568 | 3.614 | .000 | 2.587 | 8.749 |
| +1SD | 2.240 | 1.709 | 1.311 | .190 | −1.117 | 5.597 |

Note: LLCI = low level of confidence intervals, ULCI = upper level of confidence intervals, bootstrap sample size = 534

line slope between high and low groups was also used to test the effect of three interactive regression effects (social support × sports motivation) on the achievement emotion of college students to determine the influence of achievement emotion × feedback literacy on physical exercise. As shown in Fig 1, for college students with low sports motivation, social support has a positive predictive effect on achievement emotion (simple slope = 0.314, t = 7.979, p = 0.000). For college students with high sports motivation, social support has a positive predictive effect on achievement emotion, but this effect

is small, and there is no significant difference (simple slope = 0.038, t = 1.014, p = 0.311). As can be seen from Fig 2, in the case of low feedback literacy, social support has a negative predictive effect on physical exercise (simple slope = −0.670, t = −0.427, p = 0.669), but this effect is not significant. In the case of high feedback literacy, the predictive effect of social support on physical exercise is positive (simple slope = 3.224, t = 3.027, p = 0.003). Given low feedback literacy, achievement emotion has a significant positive prediction effect on physical exercise (simple slope = 9.096, t = 3.923, p = 0.000). In the case of high feedback literacy, although the predictive effect of achievement emotion on physical exercise remains positive, its effect size decreases somewhat (simple slope = 2.240, t = 1.311, p = 0.190), but there is no significant difference. It should be noted that due to the lack of objective quantitative criteria, the Physical Activity Rating Scale used in this study may underestimate the physical exercise level of college students. At the same time, the Physical Activity Rating Scale fails to comprehensively cover all forms of physical exercise. These limitations may undermine the validity of the research results. Therefore, the results need to be interpreted with caution (Fig 3).

## 5. Discussion

This study examined the association between social support, achievement emotions, and physical exercise. The results of this study add to the literature on the association of achievement emotion with social support and physical exercise, while emphasizing the role of emotion in promoting physical exercise [7,34]. Special emphasis should be placed on the role of sports motivation and feedback literacy in regulating achievement emotion and physical exercise.

This study reported that social support and achievement emotion positively predicted physical exercise, and achievement emotion mediated between social support and achievement emotion. Our findings are consistent with previous research that found enjoyment of emotions mediates between teacher support and physical exercise. However, this finding contradicts other studies that suggest the influence of social support on emotions has cultural differences [15]. With the continuous development of social economy and the Internet era, the world is gradually becoming a global village, and cultural differences are gradually narrowing. For Chinese people, having more social support does not make them feel shy, which would negatively affect their mood. This study suggests that receiving more social support, including from family and friends, helps college students to engage more in physical exercise [12,14]. Achievement emotion also mediates the indirect effect between social support and physical exercise, which indicates that achievement emotion may be connected with both social support and physical exercise, thus emphasizing its important role for promoting physical exercise among college students. According to concepts of emotionalism and hedonism in the behavioral sciences, basic human needs have been met, and people now need to obtain maximum pleasure from repeated positive experiences in the environment [35]. In this respect, achievement emotion, as one of greatest sources of pleasure related to physical exercise, may play a mediating role in the influence of social support on physical exercise.

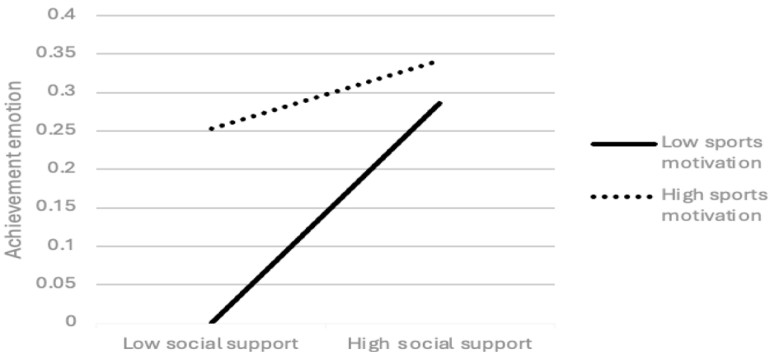

**Fig 2. Interaction effect of social support and sports motivation on achievement emotion.**

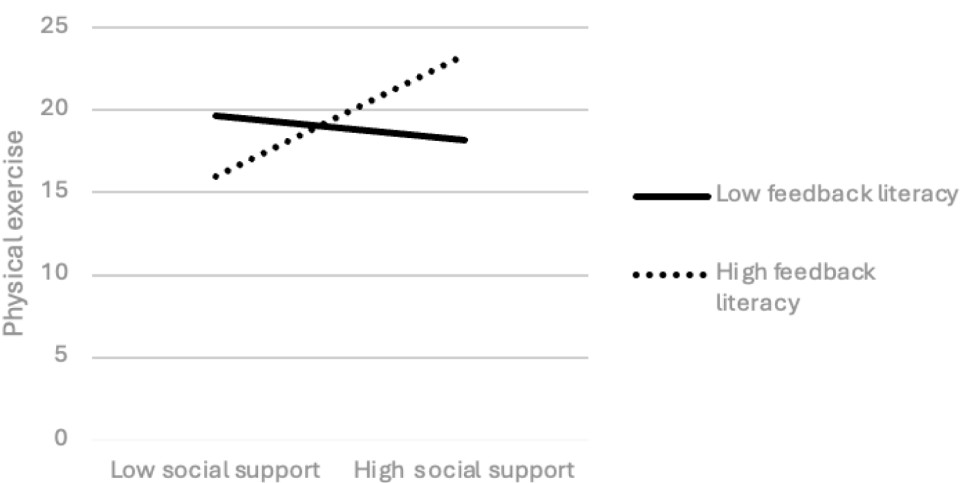

**Fig 3. Interaction effect of social support and feedback literacy on physical exercise.**

Expected value theory holds that those with higher motivation have higher expectations for things and are more likely to have higher emotions [36]. An earlier study conducted in China demonstrated that sports motivation is a crucial factor influencing emotions [37]. This suggests that higher sports motivation may lead to more positive emotions. Similarly, sports motivation generated by the setting and the achievement of short- and long-term exercise goals is positively correlated with emotional regulation [20]. Social support theory holds that social support can enhance college students' self-efficacy and thus trigger their motivation to exercise [38]. A study from China has indicated that social support has an inhibitory or preventive effect on sports motivation [39]. Colleagues' support for exercise can also increase sports motivation and predict physical exercise [40]. Despite this, few studies have directly examined the mediating role of sports motivation between social support and the promotion of mood. Only Ishii [18] have demonstrated the correlation between social support, motivation, and emotion, and their findings are consistent with those of this study, which found a positive correlation between social support, sports motivation, and achievement emotion, and that sports motivation moderated the relationship between social support and achievement emotion. Notably, this regulatory effect was reduced at higher levels of sports motivation. Increasing exercise social support for college students with stimulating their sports motivation thus needs to be maintained at a certain level to regulate their achievement emotions. However, unlike in the West, the role of sports motivation between social support and achievement emotions may be driven by a sense of collective responsibility. Therefore, attention should be paid to the application of the results.

Feedback helps individuals and groups make effective decisions [21]. For physical exercise in particular, effective and positive feedback can increase the levels of exercise among college students [25]. Previous studies have not only proved that feedback plays an important role in college students' physical exercise but also tested the positive prediction of corporate executives' physical exercise [26], and their findings are consistent with information processing theory. However, feedback literacy has received scant attention in relation to physical exercise. This study demonstrated a positive correlation between feedback literacy and physical exercise, and feedback itself is a part of social support. Positive feedback from family, friends, or others through language, goods, and behaviors is conducive to enhancing college students' engagement with physical exercise [27]. This is basically consistent with our finding that social support positively predicts physical exercise through feedback literacy in general. Of course, the relationship between emotion and feedback has long been demonstrated [28]. Positive feedback can produce high emotional states, while negative emotions can produce low emotional states [41,42]. This study has also shown that achievement emotion positively predicts physical exercise through feedback literacy. Among college students with high feedback literacy, the effect of high achievement emotion on

physical exercise is less satisfactory, which suggests that college students with different levels of feedback literacy should adjust their achievement emotions to produce an effect on physical exercise.

## 6. Conclusions

This study provides basic data for the formulation of college students' achievement emotion and physical exercise intervention strategies. It also provides support for students as they actively seek to overcome difficulties and participate in physical exercise. The findings suggest that social support from family, friends, and others can improve physical exercise levels in college students directly, as well as through positive achievement emotions. Sports motivation also has a direct and indirect adjustment effect in the front, and feedback literacy has a direct and indirect adjustment effect in the back. Special attention should therefore be paid to sports motivation and feedback literacy in this model.

Promoting the level of physical exercise among college students remains important, and it is necessary to continue to strengthen their social support. Increasing their sports motivation would be conducive to improving their self-confidence, but excessively high sports motivation is not beneficial. The feedback literacy of college students should also be improved, as this would support their levels of physical exercise, but students with very strong feedback literacy are also not supported in this model. Finally, it is necessary to pay attention to students' achievement emotion, as high achievement emotion is the key to improving their level of physical exercise. Although the theories and scales used in this study were mainly developed in a Western context, cultural factors need to be considered when applying them to the Chinese population. The collectivist norms in China may amplify the roles of social support, sports motivation, and achievement emotions in motivating physical exercise. Our study explained the influence of social support on college students' physical exercise and explored the related psychological mechanisms to provide a reference for supporting college students' physical exercise behavior.

## 7. Research limitations and suggestions for future research

There are several limitations to this study. First, the participants were recruited through convenience sampling, from only two universities in China, which limits the generalizability of the results. Second, the study did not use a longitudinal design, preventing the establishment of causal relationships. Additionally, we did not explore differences related to gender, age, or other demographic factors, which further restricts the applicability of the findings. The measurement of physical exercise relied on self-reporting; future studies should use sensors to measure physical exercise more objectively. Furthermore, expanding the sample size could facilitate a more comprehensive examination of gender and age effects. Finally, future research should investigate other potential psychological mechanisms that may mediate the impact of social support on physical exercise.

## Acknowledgments

We thank LetPub (www.letpub.com) for its linguistic assistance during the preparation of this manuscript.

## Author contributions

**Conceptualization:** Hui Zhang, Yuzhong Zhang, Baichao Xu.

**Data curation:** Hui Zhang, Yi Zhou.

**Funding acquisition:** Baichao Xu.

**Investigation:** Yuzhong Zhang, Yi Zhou.

**Methodology:** Yuzhong Zhang.

**Supervision:** Baichao Xu.

**Validation:** Yuzhong Zhang.

**Writing – original draft:** Hui Zhang, Baichao Xu.

**Writing – review & editing:** Yi Zhou, Baichao Xu.

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
