## [Decision Letter · Decision Letter 0]

11 Mar 2025

Dear Dr. zhang,

Thank you for submitting your manuscript to PLOS ONE. After careful consideration, we feel that it has merit but does not fully meet PLOS ONE’s publication criteria as it currently stands. Therefore, we invite you to submit a revised version of the manuscript that addresses the points raised during the review process.

We look forward to receiving your revised manuscript.

Kind regards,

Henri Tilga, PhD

Academic Editor

PLOS ONE

 [Hainan Provincial Key Laboratory of Sports and Health Promotion Project under Grant [number HNYJ2023005]. 

Reviewers' comments:

Reviewer's Responses to Questions

**Comments to the Author**

1. Is the manuscript technically sound, and do the data support the conclusions?

Reviewer #1: Partly

2. Has the statistical analysis been performed appropriately and rigorously?

Reviewer #1: Yes

3. Have the authors made all data underlying the findings in their manuscript fully available?

Reviewer #1: Yes

4. Is the manuscript presented in an intelligible fashion and written in standard English?

Reviewer #1: Yes

Reviewer #1: Relevant problems that justify the selection of the variables in this study and the research design chosen must be provided more substantially.

Explanations and justifications should be provided for the sampling method and instruments used.

Practical implications of the study with clear links to the findings, and discussions of how the study could contribute to the body of research in the area should be more extensive and comprehensive.

**Do you want your identity to be public for this peer review?** For information about this choice, including consent withdrawal, please see our Privacy Policy

Reviewer #1: No

---

## [Author Response · Author response to Decision Letter 1]

31 Mar 2025

Dear editor/reviewers:

Thank you for your constructive feedback and the opportunity to revise our manuscript. We have carefully addressed all comments and made corresponding revisions to the manuscript. Below is our point-by-point response to the reviewers’ concerns.

1. PLOS ONE's style Statement

Comment: Please ensure that your manuscript meets PLOS ONE's style requirements.

Response:

We have revised the manuscript according to the style requirements of PLOS ONE, please review it.

2. Role of Funder Statement

Comment: Please state the role of the funders in the study.

Response:

As requested, we have added the following statement in the cover letter:

3. Data Availability Statement

Comment: Confirm whether all raw data required to replicate the results are provided.

Response:

Raw data for all statistical analyses are provided in S1 Achievement emotion in social support and physical exercise Dataset.

https://datadryad.org/dataset/doi:10.5061/dryad.s7h44j1jh

4. Ethics Statement

Comment: Provide full ethics committee details and informed consent information.

Response:

We have added the following ethics statement to the “Methods” section:

This study was approved by the Project Approval Committee Ethics Approval Panel. Written informed consent was obtained from all participants prior to data collection.

Change in manuscript: See “Methods” section (Page 5, Line 104-105).

5.Reference List

Comment: Ensure references are complete and avoid citing retracted articles.

Response:

We have thoroughly reviewed the reference list and removed citations to any retracted articles. Verified the accuracy of all references using Google Scholar. Ensured that all cited works are formatted according to the journal’s guidelines.

We have changed references 1, 6 and 9. Because they can only look it up in China National Knowledge Infrastructure, not in Google Scholar.

Change in manuscript: Updated references are marked in the revised manuscript (Pages 19–21).

We sincerely appreciate your time and consideration. Please contact us if further revisions are required.

Best regards,

Zhang Hui

---

## [Decision Letter · Decision Letter 1]

24 Apr 2025

Dear Dr. zhang,

Thank you for submitting your manuscript to PLOS ONE. After careful consideration, we feel that it has merit but does not fully meet PLOS ONE’s publication criteria as it currently stands. Therefore, we invite you to submit a revised version of the manuscript that addresses the points raised during the review process.

We look forward to receiving your revised manuscript.

Kind regards,

Henri Tilga, PhD

Academic Editor

PLOS ONE

Reviewers' comments:

Reviewer's Responses to Questions

**Comments to the Author**

Reviewer #2: (No Response)

Reviewer #3: (No Response)

2. Is the manuscript technically sound, and do the data support the conclusions?

Reviewer #2: Partly

Reviewer #3: (No Response)

3. Has the statistical analysis been performed appropriately and rigorously?

Reviewer #2: I Don't Know

Reviewer #3: (No Response)

4. Have the authors made all data underlying the findings in their manuscript fully available?

Reviewer #2: Yes

Reviewer #3: (No Response)

5. Is the manuscript presented in an intelligible fashion and written in standard English?

Reviewer #2: Yes

Reviewer #3: (No Response)

Reviewer #2: I found this article interesting, but there are still some issues that need to be discussed with the author:

1. The research background in the first paragraph is too long, please simplify it; however, in the second paragraph, I personally think that the author should let the readers be clear about the significance of the research in this paper, please expand this part of the significance of the research.

2. The author goes directly from “Introduction” to “Methods”, I think the author should add “Literature review and hypotheses establishment”, such as explaining the relationship between the four variables that you mentioned, these relationships are not created out of thin air, he needs to be supported by theory and literature.

3. In Table 1, the first four variables are mean values, but the fifth variable, Physical exercise, seems to be a sum, whether its standard deviation is also a problem, please check the data carefully.

4. The title of the author's article discusses “moderating effect”, but why does Table 2 show “mediating effect”? Please carefully identify these two concepts to avoid confusion. In addition, please draw a diagram of the mediation or moderation model between several variables to make it easier for the reader to read and understand. This will make the subsequent data results, discussion, and conclusions clearer. The current author's writing confuses me effectively to address these suggestions.

5. Also, please reconfirm the format of the references section, for example, some citations have all capitalized authors and some have not; and is the 9th reference incomplete?

Reviewer #3: Overall, I find this study to be a valuable contribution to the field. It tackles a socially relevant issue with innovative methodology, demonstrates a substantial amount of work, and presents a clear and well-organized structure. The variables are supported by established scales, lending credibility to the findings. I believe this paper will be a meaningful addition to the literature. I have the following suggestions for revision:

Specific Comments:

Lack of Clear Research Motivation and Theoretical Context: While the authors provide specific theoretical and empirical support in the introduction, the research motivation and the current state of the relevant theory are not clearly articulated.

Potential Misinterpretation of Correlation (L168-173): The phrase “a significant positive correlation between them” could be misinterpreted by readers as implying causation. I suggest revising this to “a significant positive association between them” to more accurately reflect the nature of cross-sectional research and avoid suggesting a causal relationship.

Missing Information on Sampling Process (L178-188): This section lacks details on the sample selection process, including the timing of questionnaire administration, the sampling procedure, the calculation of the minimum sample size, and the specific distribution of the sample. This information is crucial for evaluating the scientific rigor of the study and the reliability of the results, but it is not presented in the manuscript.

Lack of Discussion on Measurement Tool Limitations (L223-243): The limitations of the measurement tool are not addressed. The text does not mention the limitations of the PARS-3 scale in terms of its scope of measurement, nor does it discuss the potential impact of these limitations on the study’s results.

Ambiguity between “Physical Exercise” and “Physical Activity” in Discussion: The terms “physical exercise” and “physical activity” are used interchangeably in the discussion. While these terms can be used synonymously in some contexts, it is important to clearly distinguish between them, particularly when selecting measurement tools, to ensure the accuracy of the study.

Lack of Discussion on Theoretical Adaptation to Chinese Context (L275-291): The manuscript mentions social support theory and expectancy-value theory but does not discuss their applicability within the Chinese cultural context. The influence of social support and motivation on affect may differ across cultures, and this warrants further discussion.

Unclear Definition and Measurement of “Feedback Literacy”: The concept of “feedback literacy” is mentioned but not clearly defined or operationalized. As a relatively new concept, it requires further elaboration on its specific meaning and measurement within the context of physical exercise.

Lack of Discussion on Cultural Adaptation of Theory and Scales (L322-332): The cultural adaptation of the theories and scales used in the Chinese context is not discussed. The impact of social support, motivation, and affect on physical exercise may vary across different cultural backgrounds.

Standardized Abstract: I recommend using a standardized abstract format, including sections on Purpose, Methods, Results, and Conclusions.

**Do you want your identity to be public for this peer review?** For information about this choice, including consent withdrawal, please see our Privacy Policy

Reviewer #2: No

Reviewer #3: No

---

## [Author Response · Author response to Decision Letter 2]

3 May 2025

Dear editor/reviewers:

Thank you for your constructive feedback and the opportunity to revise our manuscript. We have carefully addressed all comments and made corresponding revisions to the manuscript. Below is our point-by-point response to the reviewers’ concerns.

Responses to Reviewer #2

1. Background

Comment: The first paragraph’s background is too lengthy; simplify it. Expand the significance of the research in the second paragraph.

Response:

We have already streamlined the first paragraph (see the Introduction, pages 3, lines 53-69). The significance of the research has been added in the Preface (page 4, lines 84-88).

2. Literature review and hypotheses establishment

Comment: Increase “Literature review and hypotheses establishment”.

Response:

We have added “Literature review and hypotheses establishment”(page 4-8, lines 89-174).

  3. Table 1: Physical exercise

Comment: In Table 1, the first four variables are mean values, but the fifth variable, Physical exercise, seems to be a sum, whether its standard deviation is also a problem, please check the data carefully.

Response:

We used the Chinese version of the Physical Activity Rating Scale-3 (PARS-3) developed by Liang Qingde to assess physical exercise. The scale included four items rated on a 1–5 scale. Exercise volume was calculated as:

Exercise Volume=Intensity×(Time−1)×Frequency

Exercise volume categories were defined as follows:

Small exercise volume (≤19 points)

Moderate exercise volume (20–42 points)

Large exercise volume (≥43 points)

Descriptive statistics for these categories are presented in Table 1.

4. Moderating effect and Mediating effect

Comment: The title of the author's article discusses “moderating effect”, but why does Table 2 show “mediating effect”? Please carefully identify these two concepts to avoid confusion. 

Response:

To avoid ambiguity, we modify the title to: Social Support and Physical Exercise: The Mediating Role of Achievement Emotions, and the Moderating Role of Exercise Motivation and Feedback Literacy. We draw a diagram of the mediation or moderation model (page 8, lines 174).

5. References

Comment: Also, please reconfirm the format of the references section, for example, some citations have all capitalized authors and some have not; and is the 9th reference incomplete?

Response:

We have modified the capitalization of the references (page 22-28) and corrected the incorrect reference in item 9 (page 23, lines 469-470).

Responses to Reviewer #3

1. Background

Comment: Lack of Clear Research Motivation and Theoretical Context.

Response:

We have added “Literature review and hypotheses establishment”(page 4-8, lines 89-174).

2. Potential Misinterpretation of Correlation

Comment: The phrase “a significant positive correlation between them” could be misinterpreted by readers as implying causation. I suggest revising this to “a significant positive association between them” to more accurately reflect the nature of cross-sectional research and avoid suggesting a causal relationship.

Response:

We have modified "a significant positive correlation between them" to "a significant positive association between them"(page 12, lines 249-250).

3. Missing Information on Sampling Process (L178-188)

Comment: This section lacks details on the sample selection process, including the timing of questionnaire administration, the sampling procedure, the calculation of the minimum sample size, and the specific distribution of the sample. 

Response:

The above content is reflected in "Participants" (page 8-9, lines 178-195). We have also added the calculation method of the minimum sample size (page 9, lines 188-189).

4. Lack of Discussion on Measurement Tool Limitations

Comment: The limitations of the measurement tool are not addressed. The text does not mention the limitations of the PARS-3 scale in terms of its scope of measurement, nor does it discuss the potential impact of these limitations on the study’s results.

Response:

We added the limitations of the Physical Exercise Rating Scale and its impact on the research results (page 16-17, lines 313-318).

5. Ambiguity between “Physical Exercise” and “Physical Activity” in

Discussion

Comment: The terms “physical exercise” and “physical activity” are used interchangeably in the discussion. While these terms can be used synonymously in some contexts, it is important to clearly distinguish between them, particularly when selecting measurement tools, to ensure the accuracy of the study.

Response:

When we asked the relevant translation agency to translate, we confused the concepts of "physical exercise" and "physical activity". We have unified the concept of the full text (physical exercise).

6.Lack of Discussion on Theoretical Adaptation to Chinese Context

(L275-291)

Comment: The manuscript mentions social support theory and expectancy-value theory but does not discuss their applicability within the Chinese cultural context. The influence of social support and motivation on affect may differ across cultures, and this warrants further discussion.

Response:

We have added a discussion in the context of Chinese culture (page18-19, lines 351-353,357-359,369-372).

7.Unclear Definition and Measurement of “Feedback Literacy”

Comment: The concept of “feedback literacy” is mentioned but not clearly defined or operationalized. As a relatively new concept, it requires further elaboration on its specific meaning and measurement within the context of physical exercise.

Response: We have added the definition of feedback literacy (page 7, lines 145-146).

8.Lack of Discussion on Cultural Adaptation of Theory and Scales

(L322-332)

Comment: The cultural adaptation of the theories and scales used in the Chinese context is not discussed. The impact of social support, motivation, and affect on physical exercise may vary across different cultural backgrounds.

Response: We have added a discussion on "The cultural adaptation of the theories and scales used in the Chinese context"(page 21, lines 411-414).

9.Standardized Abstract

Comment: I recommend using a standardized abstract format, including sections on Purpose, Methods, Results, and Conclusions.

Response: We have completed the modification of a standardized abstract format (page 2, lines 27,33,36,45).

We sincerely appreciate your time and consideration. Please contact us if further revisions are required.

Best regards,

Zhang Hui

---

## [Decision Letter · Decision Letter 2]

13 May 2025

Social Support and Physical Exercise: The Mediating Role of Achievement Emotions, and the Moderating Role of Exercise Motivation and Feedback Literacy

PONE-D-24-55439R2

Dear Dr. zhang,

We’re pleased to inform you that your manuscript has been judged scientifically suitable for publication and will be formally accepted for publication once it meets all outstanding technical requirements.

Kind regards,

Henri Tilga, PhD

Academic Editor

PLOS ONE

Additional Editor Comments (optional):

Reviewers' comments:

Reviewer's Responses to Questions

**Comments to the Author**

Reviewer #2: All comments have been addressed

Reviewer #3: All comments have been addressed

2. Is the manuscript technically sound, and do the data support the conclusions?

Reviewer #2: Yes

Reviewer #3: Yes

3. Has the statistical analysis been performed appropriately and rigorously?

Reviewer #2: Yes

Reviewer #3: Yes

4. Have the authors made all data underlying the findings in their manuscript fully available?

Reviewer #2: Yes

Reviewer #3: Yes

5. Is the manuscript presented in an intelligible fashion and written in standard English?

Reviewer #2: Yes

Reviewer #3: Yes

Reviewer #2: (No Response)

Reviewer #3: (No Response)

**Do you want your identity to be public for this peer review?** For information about this choice, including consent withdrawal, please see our Privacy Policy

Reviewer #2: No

Reviewer #3: **Yes: ** Bo Li

---

## [Editor Report · Acceptance letter]

PONE-D-24-55439R2

PLOS ONE

Dear Dr. zhang,

I'm pleased to inform you that your manuscript has been deemed suitable for publication in PLOS ONE. Congratulations! Your manuscript is now being handed over to our production team.

Kind regards,

on behalf of

Dr. Henri Tilga

Academic Editor

PLOS ONE